# Characteristics of Gut Microbiota in Patients with Hypertension and/or Hyperlipidemia: A Cross-Sectional Study on Rural Residents in Xinxiang County, Henan Province

**DOI:** 10.3390/microorganisms7100399

**Published:** 2019-09-26

**Authors:** Huijun Li, Bingdong Liu, Jie Song, Zhen An, Xiang Zeng, Juan Li, Jing Jiang, Liwei Xie, Weidong Wu

**Affiliations:** 1School of Public Health, Xinxiang Medical University, Xinxiang 453003, China; lihuijunle@163.com (H.L.); songjie231@126.com (J.S.); azy1985@163.com (Z.A.); zengxiang3044@163.com (X.Z.); xiaoen222000@163.com (J.L.); jiangjing2019@sina.com (J.J.); 2State Key Laboratory of Applied Microbiology Southern China, Guangdong Provincial Key Laboratory of Microbial Culture Collection and Application, Guangdong Open Laboratory of Applied Microbiology, Guangdong Institute of Microbiology, Guangdong Academy of Sciences, Guangzhou 510070, China; liubingdong@stu2016.jnu.edu.cn

**Keywords:** gut microbiota, hypertension, hyperlipidemia, rural residents, central China

## Abstract

Human gut microbiota can be affected by a variety of factors, including geography. This study aimed to clarify the regional specific characteristics of gut microbiota in rural residents of Xinxiang county, Henan province, with hypertension and hyperlipidemia and evaluate the association of specific gut microbiota with hypertension and hyperlipidemia clinical indices. To identify the gut microbes, 16S rRNA gene sequencing was used and a random forest disease classifier was constructed to discriminate between the gut microbiota in hypertension, hyperlipidemia, and the healthy control. Patients with hypertension and hyperlipidemia presented with marked gut microbiota dysbiosis compared to the healthy control. The gut microbiota related to hypertension and hyperlipidemia may consist of a large number of taxa, influencing each other in a complex metabolic network. Examining the top 35 genera in each group showed that *Lactococcus*, *Alistipes*, or *Subdoligranulum* abundances were positively correlated with systolic blood pressure (SBP) or diastolic blood pressure (DBP) in hypertensive patients with treatment-naive hypertension (*n* = 63). In hypertensive patients undergoing anti-hypertensive treatment (*n* = 104), the abundance of *Megasphaera* or *Megamonas* was positively correlated to SBP. In the hyperlipidemia group, some of the top 35 genera were significantly correlated to triglyceride, total cholesterol, and fasting blood glucose levels. This study analyzed the characteristics of the gut microbiota in patients with hypertension and/or hyperlipidemia, providing a theoretical basis for the prevention and control of hypertension and hyperlipidemia in this region.

## 1. Introduction

Hypertension is a very common chronic medical condition, historically defined by sustained systolic blood pressure (SBP) above 140 mmHg or diastolic blood pressure (DBP) above 90 mmHg. Despite a variety of community-based efforts to influence lifestyle and recent pharmacological and surgical advances in the management of hypertension, its prevalence has doubled globally from 1990 to 2015 [1]. Due to the complexity and heterogeneity of hypertension, identification of the causes of this disease continues to be challenging. The homeostatic maintenance of blood pressure (BP) is a complex process, governed by the kidneys, and regulated by genetic, environmental, and endocrine factors [2,3]. In recent decades, the potential of the gut microbiome to alter host health status has drawn considerable attention. The microbiome is a microbial ecosystem that has coevolved with the host and which plays a part in the modulation of multiple physiological processes [4]. Emerging evidence suggests a link between the gut microbiome and various diseases, including atherosclerotic cardiovascular disease [5], lung disease [6], and cancer [7].

Notably, recent studies have suggested that the gut microbiome participates in BP regulation and the pathogenesis of hypertension. The rationale for examining the association between the gut microbiome and hypertension is based on the following: first, the gut is the single most important organ for adsorption of nutrients and ions, which greatly impacts BP [8]; second, dysregulation of gut microbiota is associated with various metabolic diseases and hypertension risk factors such as obesity, heart failure, and diabetes mellitus [9]; third, there is evidence of gut pathophysiology in animal hypertension models [10]; fourth, fecal transplantation from hypertension patients elevates the BP of normal animals [11]. Recently, a human metagenomic analysis of the fecal samples from 41 healthy controls, 56 subjects with pre-hypertension, and 99 subjects with primary hypertension demonstrated that aberrant gut microbiota contributes to the pathogenesis of hypertension [11]. Other studies have also reported the difference in the gut microbiome of hypertension patients and healthy controls [11,12,13]. Overall, these observations strongly indicate that the gut microbiome may play an important role in the pathogenesis of hypertension.

Human gut microbiota can be affected by a variety of factors, including geography and diet. Geography in particular exerts a strong effect on human gut microbiota [14,15]. Geographical variation should be carefully considered in case-control studies because it could manifest on small geographical scales and bias disease signals. In addition, hyperlipidemia is a major risk factor for hypertension [16,17], which could lead to changes in gut microbiota. So, it is critical to better understand how gut microbiota varies in different regions with different dietary habits, and the differences in gut microbiota between individuals with hypertension and hyperlipidemia. Henan Province is located in central China and had a population of about 108 million in 2017. It is the largest province in terms of population and agricultural production, with about half of the population living in rural areas. Thus, in this study, we endeavored to characterize the gut microbiome of residents with hypertension and hyperlipidemia from the rural area of Xinxiang county in Henan Province. In addition, the associations between gut microbes from the subjects and clinical indices or risk factors for hypertension and hyperlipidemia were analyzed in this study.

## 2. Materials and Methods

### 2.1. Study Participants

This study was approved by the Ethics Committee of Xinxiang Medical University for Human Studies (protocol number HS05, approved 20 February 2017) and all subjects gave written informed consent. To investigate the characteristics of the gut microbiota of hypertensive patients, 63 hypertensive patients with treatment-naive hypertension (NH group), 104 hypertensive patients undergoing anti-hypertensive treatment (AH group), 26 subjects with normal BP but with hyperlipidemia (HLD group), and 42 healthy subjects with normal DBP, SBP, fasting blood glucose, cholesterol, and triglyceride levels (control group) in the rural area of Henan Province, central China were randomly recruited in this study. For the hypertensive patients in the AH group, nifedipine, metoprolol, statins, diuretics and angiotensin-converting enzyme inhibitors, and angiotensin II receptor antagonists were the most commonly used drugs. Hypertensive and/or hyperlipidemic patients with other chronic diseases including cancer, heart failure, diabetes mellitus, renal failure, chronic respiratory disease, peripheral artery disease, metabolic disorders, and inflammatory bowel disease were excluded. The healthy controls had no history of chronic diseases and were not taking any medicine that might disrupt gut microbiota, such as antibiotics, probiotics, and anti-inflammatory agents. The participants were older than 18 years, met the inclusion criteria, and were competent and willing to provide written consent.

### 2.2. Study Protocol and Evaluation Criteria 

All on-site examinations were performed at the local medical stations or community clinics that usually provided clinical services for the participants from April to June 2017. Medical information was recorded according to standard procedures. BP was measured by trained medical students with an electronic sphygmomanometer (OMRON HEM-7071 professional portable blood pressure monitor, OMRON, Shanghai, China) on the right arm positioned at heart level in a seated position. The participants were required to rest for at least 5 min before BP measurement. An appropriately sized cuff was selected based on the circumference of the right upper arm of the participants to avoid overlapping. BP was measured three times during one visit, and the mean of the three readings was used in the analysis. Hypertension was defined as SBP ≥ 140 mmHg, DBP ≥ 90 mmHg, or by self-reported use of antihypertensive medications in the last 2 weeks irrespective of BP values. Levels of fasting blood glucose (FBG), glycosylated hemoglobin (HbA1c), total cholesterol (TC), triglyceride (TG), high-density lipoprotein (HDL), and low-density lipoprotein (LDL) were measured with a SIEMENS ADVIA 2400 auto analyzer (Siemens, München, Germany). In accordance with the Chinese guideline on the prevention and treatment of hyperlipidemia in adults, hypercholesterolemia was defined as TC > 5.72 mmol/L, high LDL as LDL > 3.64 mmol/L, hypertriglyceridemia as TG > 1.70 mmol/L and low HDL as HDL < 0.91 mmol/L. Hyperlipidemia was defined as the presence of one or more abnormal serum lipid concentrations. Waist circumference (WC), height, and body weight were measured according to the standard protocols. Body mass index (BMI) was calculated as weight in kilograms divided by height in meters squared. The validated interview questionnaire [17] was conducted via face-to-face interviews by trained medical students.

### 2.3. Fecal Sample Collection and 16S Ribosomal RNA Sequencing

Fresh fecal samples collected from each participant were immediately frozen at −20 °C, transported to the laboratory and then stored at −80 °C until further analysis. Bacterial DNA was extracted at Beijing Novogene Bioinformatics Technology Co., Ltd. using the Cetyltrimethyl Ammonium Bromide/Sodium Dodecyl Sulfonate method. The V3–V4 regions of the 16S rRNA genes were amplified using specific primers (341F: 5′-CCTAYGGGRBGCASCAG-3′; 806R: 5′-GGACTACNNGGGTATCTAAT-3′). All PCR reactions were carried out with Phusion^®^ High-Fidelity PCR Master Mix (Beijing, New England Biolabs). Sequencing libraries were generated using the Ion Plus Fragment Library Kit 48 rxns (Thermo Scientific, Shanghai, China) following the manufacturer’s instructions. Library quality was assessed on a Qubit^®^ 2.0 Fluorometer (Thermo Scientific, Shanghai, China). Finally, the library was sequenced on an Ion S5™ XL platform and 600 bp single-end reads were generated.

### 2.4. Analysis of 16S Ribosomal RNA Sequencing Data

Single-end reads were assigned to samples based on their unique barcode and truncated by cutting off the barcode and primer sequence. The raw reads were filtered with conditions to obtain high-quality clean reads according to the Cutadapt quality controlled process. The reads were compared to the reference database using the UCHIME algorithm [18] to detect and remove chimeric sequences [19], and obtain clean reads. Sequences analysis was performed with Uparse [20]. Sequences with a similarity ≥97% were assigned to the same OTUs. Representative sequences for each OTU were screened for further annotation.

### 2.5. Statistical Analysis 

Continuous data are presented as mean ± SD. Categorical variables are presented as proportions. Data entry and management were performed using SPSS 22.0 and all *p* values are from two-tailed tests. A *p* value of less than 0.05 in a two-tail test was considered to be statistically significant. Statistical analyses were conducted using the R platform. Distance-based redundancy analysis was performed on normalized taxa abundance matrices with the R vegan package according to the Bray-Curtis distance. Random forest models were trained with the R random Forest package (10,000 trees) to predict hypertension status according to genera abundance profiles. The performance of the predictive model was evaluated with cross-validation error. Receiver operator characteristic (ROC) analysis was performed using the R *pROC* package. The *p*-value was calculated to evaluate the false discovery rate and was corrected for multiple comparisons. We analyzed the Spearman’s correlation between the top 35 enriched gut bacteria genera in the NH, AH, HLD and control subjects and clinical indices of hypertension at the genus level. The *p* values were corrected for multiple testing with the Holm method by R vegan.

## 3. Results

### 3.1. Demographic Characteristics and Clinical Indices of the Study Population

As described above, 234 rural residents in Xinxiang county, Central China, were recruited to identify the differences in gut microbiome between the NH, AH, HLD, and control groups. The demographic characteristics and clinical indices of the four groups of subjects are shown in Table 1. Members of the NH and AH groups had significantly higher BMIs than those in the control group (*p* = 0.019 and 0.027 respectively). There were significant differences in both DBP and SBP between the NH and HLD groups, NH group and control, AH and HLD groups, and AH group and control (*p* = 0.000). The HLD group had significantly higher levels of FBG than the NH, AH, and control groups (*p* = 0.004, 0.001, and 0.000, respectively). The levels of TG in the NH, AH, and HLD groups were all significantly higher than that in the control group (*p* = 0.001, 0.002, and 0.016, respectively). The levels of LDL and WC have the same trend (*p* = 0.025, 0.036, 0.000; *p* = 0.011, 0.004, and 0.048, respectively).

### 3.2. Distribution of Abundant Gut Microbiota

We acquired a total of 20,235,921 raw reads. After filtering to obtain the high-quality clean reads in accordance with the Cutadapt (V1.9.1, Department of Computer Science, TU Dortmund, Germany) quality controlled process, 18,785,481 high-quality clean reads were obtained. Sequences with ≥97% similarity were clustered into the same OTU, the representative sequence for each OTU was screened for further annotation. Taxonomically, 18 bacterial phyla and 789 bacterial genera were detected in the four groups. The top 10 phyla and genera present in the four groups are listed in Figure 1a,b, respectively.

As depicted in Figure 1a, Firmicutes, Bacteroidetes, Proteobacteria, and Actinobacteria were the most abundant bacteria phyla in all four groups. At the genus level, as shown in Figure 1b, the dominant bacterial genera in the NH group were *Blautia* (8.68%), *Bacteroides* (7.74%), and *Faecalibacterium* (6.47%). Contrastingly, in the AH group, the dominant bacteria genus was *Prevotella 9* (9.40%), which was more abundant than in the other three groups. Compared with the NH group, the relative abundance of *Blautia* (7.81%) was lower, however, the relative abundance of *Faecalibacterium* (8.76%) was higher. In HLD group, the dominant bacterial genera were *Bacteroides* (11.77%) and *Faecalibacterium* (10.50%), were more abundant than in the NH and AH groups. In the control group, the dominant bacterial genera were *Bacteroides* (9.81%), and *Faecalibacterium* (9.84%), which were the same genera seen in the HLD group, however, their relative abundances were lower than in the HLD group, but higher than in the NH group. The differential enrichment of gut microbiota in the hypertension and control groups demonstrated the dysbiosis of gut microbial in hypertension.

### 3.3. Bacteria Genera-Based Classification of Hypertension and Hyperlipidemia

To explore the genera contributing to the differences between pairs of the four groups, we constructed a random forest disease classifier using the samples from the AH, NH, HLD, and control groups. Fivefold cross-validation was repeated five times and ROC curves for distinguishing any pair of the four groups were developed.

As shown in Figure 2, the AUCs of the classifiers discriminating between the pairs were AH vs. HLD (0.831, 95% CI: 0.743–0.919), NH vs. control (0.823, 95% CI: 0.732–0.914), AH vs. control (0.776, 95% CI: 0.683–0.870), HLD vs. control (0.756, 95% CI: 0.606–0.906), AH vs. NH (0.708, 95% CI: 0.615–0.800), NH vs. HLD (0.686, 95% CI: 0.551–0.822) (Figure 2a,b). This indicated that bacterial genera were most effective for differentiating between the AH and HLD groups (AUC, 0.831). However, we observed poor performance on the training set when discriminating between NH and HLD (AUC, 0.686) which showed lower specificity and sensitivity.

Of the selected genera, *Tyzzerella*, *Flavonifractor*, *Faecalitalea*, *Dorea*, and *Phascolarctobacterium* were the common contributors for distinguishing the control group from both the AH and NH groups (Figure 3a,b). *Pediococcus, Bacteroides, Lachnospira,* and *Dorea* were the common contributors for distinguishing the control group from both the HLD and NH groups (Figure 3c,b). *Caproiciproducens*, *Mitsuokella*, *Adlercreutzia*, *Catenibacterium*, unidentified *Clostridiales*, *Hypnocyclicus*, and *Marvinbryantia* were the common contributors for distinguishing the HLD group from both the AH and NH groups (Figure 3d,e). *Butyricicoccus*, *Chitinibacter*, *Subdoligranulim*, *Thiobacillus*, and *Tyzzerella* all contributed to distinguishing the AH from the NH group (Figure 3e).

### 3.4. Association of Gut Microbiome with Clinical Indices of Hypertension and Hyperlipidemia

To further determine the clinical implications of alterations in bacterial genera abundance in the pathogenesis of hypertension, Spearman’s correlation analysis was used to examine the correlations between bacterial genera abundance, clinical indices, and risk factors for hypertension (SBP, DBP, BMI, FBG, HbA1c, TC, TG, age, WC, HDL, and LDL). Fecal microbial genera were significantly associated with many clinical indices of the hypertension tested.

In NH group, some of the top 35 genera were significantly correlated with age, BMI, TC, HDL, DBP, SBP, and WC. Of these, the abundances of *Lactococcus*, *Alistipes*, or *Subdoligranulum* were positively correlated to SBP and DBP (Figure 4a). In the AH group, some of the top 35 genera were significantly correlated to the clinical indices and risk factors of hypertension except for HDL. Notably, the abundance of *Megasphaera* and *Megamonas* was positively correlated to SBP, while the abundances of *Clostridium sensu stricto 1*, *Romboutsia*, *Erysipelotrichaceae UCG.003*, *Ruminococcus 2*, and *Intestinibacter* were negatively correlated to SBP and DBP. *Ruminococcus 2*, was also positively correlated to age, TC, and LDL (Figure 4b). In the HLD group, some of the top 35 genera were significantly correlated to the clinical indices and risk factors other than age and SBP, especially TG, TC, FBG, and LDL. *Alistipes*, *Intestinibacter*, *Subdoligranulum*, and unidentified *Ruminococcaceae* were significantly negatively correlated to TG. The abundance of *Megasphaera* was significantly positively correlated to FBG, while the abundances of unidentified *Lachnospiraceae*, *Romboutsia*, and *Bifidobacterium* were significantly negatively correlated to FBG (Figure 4c). In control group, some of the top 35 genera significantly correlated to the clinical indices and risk factors except for FBG. The abundance of the *Christensenellaceae* R.7 group, *Alistipes*, *Parabacteroides*, *Escherichia*, and *Shigella* were negatively correlated to SBP and DBP. In addition, the *Christensenellaceae* R.7 group and *Alistipes* were also negatively correlated to WC and BMI. Meanwhile, the abundances of *Pseudomonas*, *Fusicatenibacter*, *Tyzzerella 3*, *Lactococcus*, *Veillonella*, and *Megamonas* were positively correlated to SBP or DBP (Figure 4d). We could see that a large number of gut microbiota taxa may be related to hypertension, influencing each other in a complex metabolic network, rather than being solely due to a single player or a limited number of species.

## 4. Discussion

The pathogenesis of hypertension is complex and, in most cases, multifactorial. Evidence in recent years has implicated gut microbiota in hypertension [21,22,23]. This notion was supported by the results of this study, in which we showed that hypertension patients, regardless of whether they received anti-hypertensive treatments, displayed remarkable gut microbiota dysbiosis. Studies that explore the relationships between human gut microbiota, lifestyle changes, and disease epidemiology at the regional level will provide important data for public health science.

In this study, we determined the alterations in the taxa of the gut microbiomes of patients with hypertension or hyperlipidemia and healthy subjects in rural residents of Xinxiang county, Henan province. Furthermore, this study demonstrated that correlation analysis of taxa abundance to clinical indices of hypertension could help identify gut microbes that had significant impacts on hypertension. For example, we observed that the abundances of *Lactococcus*, *Alistipes*, and *Subdoligranulum* were positively correlated with SBP and DBP in hypertensive patients not receiving anti-hypertensive treatment. *Prevotella* was found to be more abundant in the AH and HLD groups than in the NH and control groups in this study, and its abundance was negatively correlated to LDL and WC in the AH group. Previous studies have suggested that the intestinal bacterium *Prevotella copri* thrives in the pro-inflammatory environment of rheumatoid arthritis, as the superoxide reductase and phosphoadenosine phosphosulphate reductase encoded by *Prevotella copri* may favor the development of inflammation [24]. Thus, it is thought that *Prevotella* may play a critical role in the pathogenesis of hypertension through triggering the inflammatory response.

Similar to a previous observation [11], the findings of this study demonstrated that *Faecalibacterium* and *Bifidobacterium* were enriched in the healthy control and AH groups but decreased in the NH group. However, *Bifidobacterium* was shown to be positively correlated to WC in the NH group. *Bifidobacterium* is an important group of probiotic bacteria, commonly used in fermenting dairy products, that constitutes a major part of the human intestinal microbiota in healthy humans [25] and is also a major acetate-producing bacteria [26]. As for *Faecalibacterium*, its abundance is positively correlated to HDL but negatively correlated to BMI and WC in the control group. *Faecalibacterium* is a butyrate-producing bacterium [13]. Clinical evidence has indicated that the abundance of butyrate producing bacteria is associated with lower blood pressure in obese pregnant women [14]. In this study, *Faecalibacterium* contributed to differentiating the control group from the NH group (Figure 3b). *Megasphaera* was shown to be enriched in healthy controls in previous study [12]. However, in this study the abundance of *Megasphaera* was shown to be positively correlated to SBP in the AH group and positively correlated to FBG in the HLD group. Among these selected genera, *Megasphaera* contributed to distinguishing between the HLD and the control groups (Figure 3d), as well as the HLD and AH groups (Figure 3a) in a random forest model. These inconsistent results might be due to variations in diet, geographical differences, and other unknown factors.

In addition, *Parabacteroides* was found to be significantly negatively correlated to BMI and DBP in the control group (Figure 4d), and, using a random forest model, contributed to distinguishing between the NH and control groups (Figure 3b). The abundance of *Megamonas* was higher in the AH and NH groups than in the control group, and as expected, was positively correlated to SBP in the AH group (Figure 4b) and contributed to distinguishing control from the NH group (Figure 3b). Intriguingly, the abundances of *Clostridium sensu stricto 1*, *Romboutsia*, and *Faecalibacterium* were higher in the control group than in the NH or AH groups (Figure 1b). *Clostridium sensu stricto 1* and *Romboutsia* were negatively correlated to SBP or TG in the AH group (Figure 4b). *Faecalibacterium* was significantly negatively correlated to WC and BMI, and positively correlated to HLD in the control group (Figure 4b). *Faecalibacterium* and *Romboutsia* contributed to discriminating between the NH and control groups (Figure 3b). It is noteworthy that a large number of gut microbiota taxa may be related to hypertension, influencing each other in a complex metabolic network, rather than a single player or a limited number of species.

In this study, we observed poor performance on the training set when discriminating between the NH and HLD groups (AUC, 0.686) with lower specificity and sensitivity. This confirmed that subjects in the HLD and NH groups had similar gut microbiome compositions, although both DBP and SBP are significantly different in the NH, and HLD groups. Correlation analyses of hypertension-related clinical indices and gut microbiome demonstrated that serum levels of HbA1c, HDL, TG, TC, and WC, the putative risk factors for hypertension, in the subjects in the HLD group, were significantly higher than those in the control group but were comparable to those in the NH group. These findings suggest that for early BP control, attention should be paid to previously neglected populations with hyperlipidemia since these people have hypertension-related gut microbiota dysbiosis. Early gut microbiome management for individuals with hyperlipidemia may have a strong clinical implication for disease intervention.

ROC analysis of the disease classifiers using 30 bacterial genera produced AUCs for AH vs. HLD (0.831), NH vs. the control (0.823), AH vs. the control (0.776), HLD vs. the control (0.756), NH vs. AH (0.708), and HLD vs. NH (0.686). We achieved reliable discrimination between AH and HLD, NH and the control, AH and the control, and between HLD and the control. This discriminatory power is higher than that from the prediction models based on genomic markers identified by GWAS [27,28], and is at the same level as phenotype-based models (AUC 71–81%) [29,30] and gut microbiota-based models (AUC = 0.78) [12]. Since the participants in this study lived in the same area, their diets and environments are similar so that the influence of diet and the environment on the gut microbiome is reduced to a great extent. Under these circumstances, the fecal microbiota has good potential for the prediction and early diagnosis of hypertension. We noticed the relatively low AUC when discriminating between the HLD and NH groups (0.686) using bacterial genera, indicating the similar composition of the gut microbiota in hyperglycemia and hypertensive patients. The AUC when discriminating between the AH and NH groups was 0.708, indicating that administration of antihypertensive medicine may modify the composition of the gut microbiome. As depicted in Figure 1b, the relative abundance of *Bifidobacterium* in the AH group (7.55%) was much higher than in the NH group (5.18%), showing that antihypertensive medicine may increase the relative abundance of some beneficial bacteria. Medicine-induced shifts of the gut microbiome have been observed during the treatment of multiple diseases, inducing metformin therapy for type 2 diabetes [31,32] and atherosclerotic cardiovascular disease [5]. In the AH group, only 25% of the hypertensive patients’ blood pressure was effectively controlled, so it is important to understand the interaction between hypertension drugs and gut microbiota, and to explore new, more effective blood pressure control methods. In this study, most of the patients in the AH group took more than two drugs, and the types of drugs were different. It is therefore hard to determine exactly what each drug does to the gut microbiota. This study is just a preliminary exploration, and the effects of drugs will be examined in detail in future studies.

For most patients with hypertension, measuring blood pressure is a common usage for hypertension diagnosis. According to the Guideline for the prevention, detection, evaluation and management of high blood pressure in adults, patients with blood pressure ≥130 mmHg and/or diastolic pressure ≥80 mmHg are diagnosed with hypertension, and given recommendations to receive treatment [33]. However, blood pressure is not a stable but dynamic parameter from minute to minute. Previous study reported blood pressure presented a circadian rhythm during 24 h time with the highest readings in mornings and evenings and the lowest readings at night [34]. Moreover, it also fluctuates in response to temperature, noise, emotional stress, consumption of food or liquid, dietary factors, physical activity, and even in changes in posture, such as standing-up. For example, White-coat hypertension (WCH) is a familiar phenomenon and also an urgently tough problem to be solved for doctors. Due to the fear that blood pressure measurement may indicate future illness, subjects shows hypertension during the clinic visits resulting in isolated clinic hypertension [35,36]. All above mentioned strongly suggested measuring blood pressure is not an accurate approach for hypertension diagnosis.

On the other hand, as high-risk factor and common comorbidity, hyperlipidemia diagnosis is highly dependent on blood samples [37]. Although modern technology reduces the probability of infection and transfusion transmitted diseases, invasive examination remains a challenge to medical condition in most rural area. In particular, because of concerns over pain or health care, patients are reluctant to take regular blood tests. Therefore, a convenient and safe method for hyperlipidemia diagnosis is in need for people in less-developed regions.

As numerous studies proved, gut microbiota may exert a great influence on the functionality and pathophysiology of human diseases, including obesity, type II diabetes, cardiovascular diseases, and some psychiatric disorders, while a deluge of research utilized it to identity patients [38]. It was reported that gut microbes could be utilized as non-invasive biomarkers for early hepatocellular carcinoma using a random forest classifier [39]. Another research study also demonstrated Artificial Neural Network training by relative abundance of microbiota could identify insomnia patients without subjective bias in clinic practice [40]. Thus, we incorporated machine technology to identify discriminant genera and established a prediction model for hypertension and hyperlipidemia diagnosis without any disadvantages other than the ones that appeared for the previous classic method, which may provide a new perspective for the interpretation of diseases and additional auxiliary diagnosis especially for rural area in the future.

Although our study reveals the important roles of the gut microbiome in hypertension, it has several limitations. First, the sample size is limited due to the high cost of sequencing and sample availability. Study subjects were recruited from a single clinic within a small geographic area and were divided based on BP alone, without consideration for possible confounding factors such as diet, culture, or demographics. Second, the influence of different kinds of hypertension drugs on the gut microbial community in the AH group was not considered. Therefore, our data need to be interpreted with caution. Generally, hypertension is a highly complex and heterogeneous disease, it is still infeasible to draw any conclusions about the causal relationships between the gut microbiota and hypertension.

## 5. Conclusions

This study showed that a large number of gut microbiota taxa may be related to hypertension, influencing each other in a complex metabolic network, and is not solely a single player or a limited number of species. Collectively, the findings of this study expand on previous knowledge of the correlation between gut microbiota and hypertension, and provide a range of genomic biodiversity signatures for the hypertensive gut microbiome.

## Figures and Tables

**Figure 1 microorganisms-07-00399-f001:**
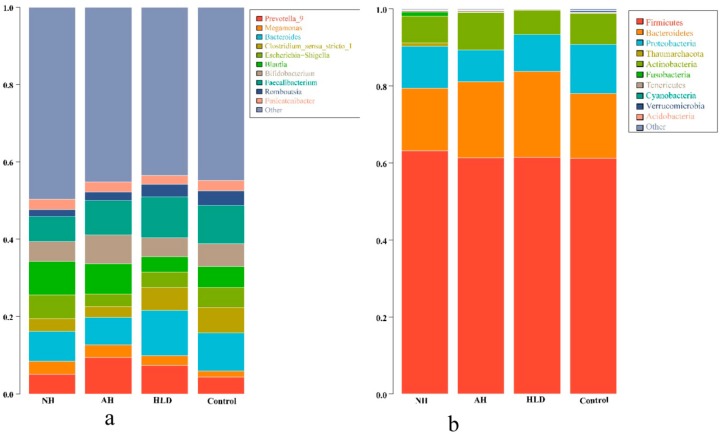
Phylogenetic profiles of gut microbes in the NH, AH, HLD, and control groups. Composition of fecal microbiota at the phylum (**a**) and genus (**b**) levels.

**Figure 2 microorganisms-07-00399-f002:**
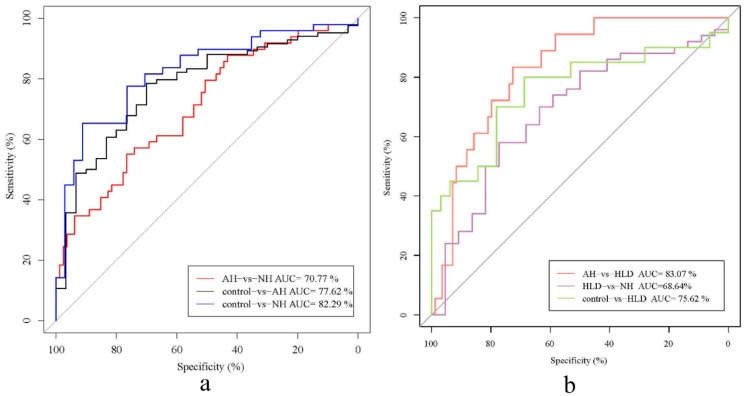
Receiver operator characteristic (ROC) analysis of the classification of patients with hypertension and hyperlipidemia the control based on the abundance of bacterial genera. (**a**) AUC for NH vs. control (blue curve), AH vs. control (black curve), and NH vs. AH (red curve). (**b**) AUC for HLD vs. NH (purple curve), HLD vs. AH (red curve), and HLD vs. control (green curve).

**Figure 3 microorganisms-07-00399-f003:**
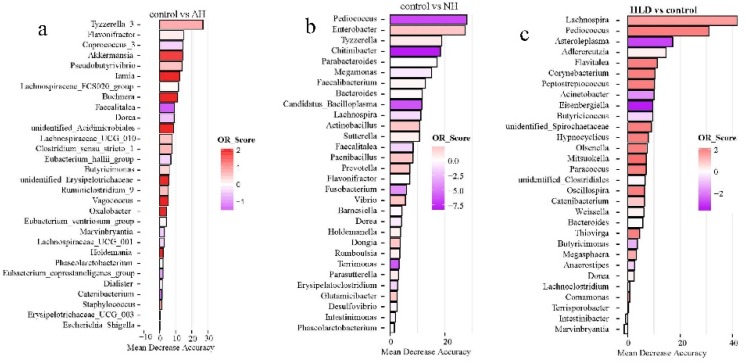
The top 30 most discriminating genera based on the random forest model discriminating between pairs of NH, AH, HLD, and control groups. (**a**) AH and control. (**b**) NH and control. (**c**) HLD and control. (**d**) HLD and AH. (**e**) HLD and NH. (**f**) AH and NH. The lengths of the bars in the histograms represent the mean decrease accuracy, which indicates the importance of the genus for classification. The color denotes the enrichment of the genus according to OR score, purple for first group and red for second group of each figure.

**Figure 4 microorganisms-07-00399-f004:**
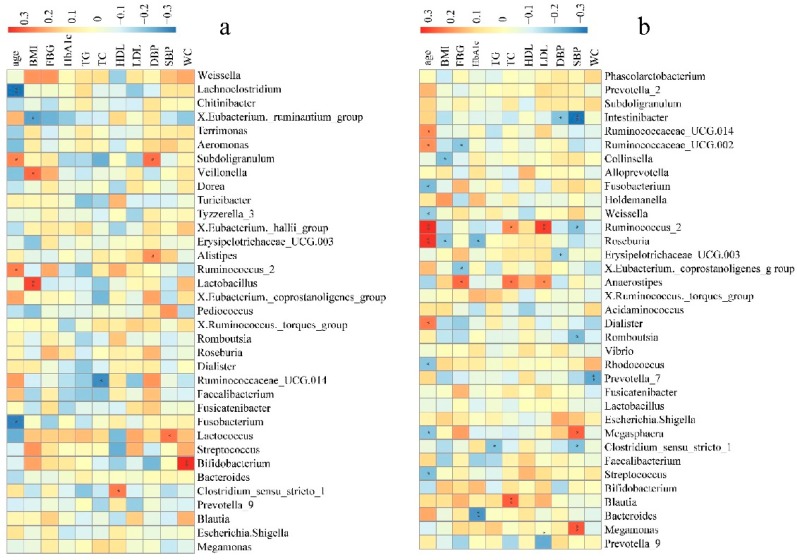
Bacterial genera correlated with clinical indices in the NH (**a**), AH (**b**), HLD (**c**), and control (**d**) groups. The color is scaled with the correlation coefficients, positive correlation is red, and negative correlation is blue. * adjusted *p* < 0.05; ** adjusted *p* < 0.01.

**Table 1 microorganisms-07-00399-t001:** Characteristics of the participants in naive hypertension (NH), anti-hypertensive (AH), hyperlipidemia (HLD), and control groups. BMI refers to body mass index, SBP refers to systolic blood pressure, DBP refers to diastolic blood pressure, FBG refers to fasting blood glucose, HDL refers to high-density lipoprotein, LDL refers to low-density lipoprotein, TG refers to triglyceride, TC refers to total cholesterol, WC refers to waist circumference.

	NH (*n* = 63)	AH (*n* = 104)	HLD (*n* = 26)	Control (*n* = 42)
Gender (Female/Male)	28/35	54/50	12/14	25/17
Age (year)	58.4 ± 10.2	59.8 ± 9.3	56.7 ± 10.0	59.3 ± 9.2
BMI (kg/m^2^)	27.0 ± 3.6 *	26.6 ± 3.1 *	26.5 ± 3.0	25.3 ± 2.9
SBP (mmHg)	92.5 ± 8.4 ***	90.7 ± 11.8 ***	78.7 ± 6.6	77.0 ± 7.6
DBP (mmHg)	149.8 ± 11.6 ***	148.8 ± 18.0 ***	126.3 ± 10.4	122.3 ± 11.5
FBG (mmol/L)	5.8 ± 1.3	5.8 ± 1.4	6.7 ± 2.2 ***	5.4 ± 0.5
HbA1c (%)	3.4 ± 0.7 *	3.4 ± 0.6 *	3.7 ± 1.3 **	3.0 ± 0.7
HDL (mmol/L)	1.3 ± 0.4	1.3 ± 0.3	1.2 ± 0.3 *	1.3 ± 0.3
LDL (mmol/L)	3.1 ± 0.8 *	3.0 ± 0.7 *	3.4 ± 0.8 ***	2.8 ± 0.5
TG (mmol/L)	2.1 ± 1.8 **	2.0 ± 1.2 **	2.2 ± 1.1 *	1.2 ± 0.4
TC (mmol/L)	5.6 ±0.9 *	5.6 ± 1.6	5.7 ± 0.9 **	5.0 ± 0.7
WC (cm)	90.8 ± 10.8 *	91.1 ± 10.0 **	90.9 ± 8.7 *	85.8 ± 9.1

*: *p* < 0.05; **: *p* < 0.01; ***: *p* < 0.001 vs. control.

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
