# Peer review of "Characteristics of Gut Microbiota in Patients with Hypertension and/or Hyperlipidemia: A Cross-Sectional Study on Rural Residents in Xinxiang County, Henan Province"

_microorganisms, 2019, doi:10.3390/microorganisms7100399_

Round 1
Reviewer 1 Report
The manuscript entitled : Characteristics of gut microbiota between hypertension and hyperlipidemia: a cross-sectional study with rural residents in Xinxiang county, Henan province by Li et al studies the gut microbiota composition in four groups two with hypertension, treated or not, one with hyperlipidemia and a healthy control group and the association with clinical markers. In general, the manuscript is very descriptive and no real conclusion is formed. The fact that Gut microbiota dysbiosis contributes to the development of hypertension has already been described in literature. The reviewer does not see the use of applying a ROC analysis to distinguish hypertension patients and hyperlipidemia from control by the abundance of bacteria genera as a simple clinical/laboratory test can make this classification. Data are nicely presented in the figures, however, it is very difficult for the reader to extract a clear message from these data.
Comments:
Abstract
Title Characteristics of gut microbiota between hypertension and hyperlipidemia: a cross-sectional study with rural residents in Xinxiang county, Henan province. Better: Characteristics of gut microbiota in patients with hypertension and/or hyperlipidemia: a cross-sectional study with rural residents in Xinxiang county, Henan province
Authors write that especially geography affects human gut microbiota. It is actually the geography related factors and especially dietary habits that do so.
Better write patients with hypertension and/or hyperlipidemia.
The activities of gut microbiota related to… In the current paper microbiota were only studied at a compositional levels not at a functional levels!
There is no conclusion in the abstract.
Introduction
What does “multipronged” mean (page 1 line 30)
Do not write first, second, third, fourth with a capital letter
Also change in line 54-55 especially geography (see comment above)
Materials and methods
Line 70: …all subject were given written informed consent. Change to …all subject gave written informed consent.
Authors mention the medication, antibiotics, probiotics and anti-inflammatory agents intake for the healthy controls but not for the patients groups, please complete.
A questionnaire is mentioned on line 102. Could the authors specify the type, is this a validated questionnaire, how was this incorporated in the data analysis?
Please write CTAB/SDS in full
Point 2.5 needs to be incorporated under 2.6 statistical analysis
Results
Please omit the first 3 instruction lines.
Under 3.1 do not describe the things that are not statistically significant, only the ones that are.
The presentation of the statistics in Table is unusual. Please mention only statistically different effects by using a symbol.
Under 3.2 line 172: …”much” higher than… , is this significantly higher?
What do the authors mean by:
line 175-176: The differential enrichment of gut microbiota in hypertension and control groups the occurrence of gut microbial dysbiosis in hypertension.
Line 182: Fivefold cross-validation was repeated for five times
The reviewer is not used to seeing AUC presented as % why not AUC (95% CI) of 0.831 (0.742 to 0.919)
The reviewer does not see the use of applying a ROC analysis to distinguish hypertension patients and hyperlipidemia from control by the abundance of bacteria genera as a simple clinical/laboratory (blood pressure, cholesterol) test can provide the classification.
Paragraph 3.4 is not readable, it contains too much information, already provided in the figure 4, and the main message is lost.
Discussion
The effect of medication intake should be better discussed
P8 line 302: what is the relevance of Prevotella, originating from mouth and vagina,…?
Author Response
Reviewer(s)' Comments to Author:
Reviewer: 1
Comments to the Author
Comment 1. Abstract
Title Characteristics of gut microbiota between hypertension and hyperlipidemia: a cross-sectional study with rural residents in Xinxiang county, Henan province. Better: Characteristics of gut microbiota in patients with hypertension and/or hyperlipidemia: a cross-sectional study with rural residents in Xinxiang county, Henan province.
Response: Thank you very much for your comments and suggestion. The title has been modified to “Characteristics of gut microbiota in patients with hypertension and/or hyperlipidemia: a cross-sectional study on rural residents in Xinxiang county, Henan province”.
Comment 2. The activities of gut microbiota related to… In the current paper microbiota were only studied at a compositional levels not at a functional levels!
Response: We appreciate the reviewer's comments. We have revised the sentence as “The gut microbiota related to hypertension and hyperlipidemia may consist of a large number of taxa, influencing each other in a complex metabolic network”. See Lines 22-24.
Comment 3. There is no conclusion in the abstract.
Response: Thank you very much for the reviewer's suggestion. We have added the conclusion statement in the abstract as “This study analyzed the characteristics of the gut microbiota in patients with hypertension and/or hyperlipidemia, providing a theoretical basis for the prevention and control of hypertension and hyperlipidemia in this region.” at the end of the abstract. See Lines 29-32.
Comment 4. Introduction
What does “multipronged” mean (page 1 line 30)
Do not write first, second, third, fourth with a capital letter
Also change in line 54-55 especially geography (see comment above)
Response: Thank you very much for your suggestion. “multipronged” means “a variety of”. We have replaced “multipronged” with “a variety of”, See Line40. And we have revised the “first, second, third, fourth”. See Lines 52-56.
Comment 5. Materials and methods
Line 70: …all subject were given written informed consent. Change to …all subject gave written informed consent.
Response: Thank you very much for the reviewer's suggestion. We have revised the sentence as “…all subject gave written informed consent”. See Lines 79-80.
Comment 6. Authors mention the medication, antibiotics, probiotics and anti-inflammatory agents intake for the healthy controls but not for the patients groups, please complete.
Response: We appreciate the reviewer's comments. We have revised the sentence as “Hypertensive and/or hyperlipidemic patients with other chronic diseases including cancer, heart failure, diabetes mellitus, renal failure, chronic respiratory disease, peripheral artery disease, metabolic disorders, and inflammatory bowel disease were excluded.” See Lines 87-90.
Comment 7. A questionnaire is mentioned on line 102. Could the authors specify the type, is this a validated questionnaire, how was this incorporated in the data analysis?
Response: Thank you very much for the reviewer's suggestion. This is a validated interview questionnaire. We used the same questionnaire which reported in the following article.
Liu X, Li Y, Guo Y, Li L, Yang K, Liu R, Mao Z, Bie R, Wang C. The burden, management rates and influencing factors of high blood pressure in a Chinese rural population: the Rurral Diabetes, Obesity and Lifestyle (RuralDiab) study[J]. J Hum Hypertens. 2018; 32(3):236-246.
Comment 8. Please write CTAB/SDS in full
Response: Thank you very much for the reviewer's suggestion. We have revised CTAB/SDS as “Cetyltrimethyl Ammonium Bromide / Sodium Dodecyl Sulfonate”. See Lines 120-121.
Comment 9. Point 2.5 needs to be incorporated under 2.6 statistical analysis
Response: Thank you very much for the reviewer's suggestion. Point 2.5 has been incorporated under 2.6 statistical analysis. See Lines 148-151.
Comment 10. Results
Please omit the first 3 instruction lines.
Under 3.1 do not describe the things that are not statistically significant, only the ones that are.
Response: Thank you very much for the reviewer's suggestion. We have deleted the contents about the indices that are not statistically significant. See Lines 158-161.
Comment 11. The presentation of the statistics in Table is unusual. Please mention only statistically different effects by using a symbol.
Response: Following the reviewer's suggestion, we have mentioned only statistically different effects by using a symbol. See Line 165, Table 1
Comment 12. Under 3.2 line 172: …”much” higher than… , is this significantly higher?
Response: Thank you very much for the reviewer's query. We have revised the sentence as “ were more abundant than in the NH and AH groups.” See Line 181-182.
Comment 13. What do the authors mean by: line 175-176: The differential enrichment of gut microbiota in hypertension and control groups the occurrence of gut microbial dysbiosis in hypertension.
Response: Thanks for the reviewer's query. We have revised the sentence as “The differential enrichment of gut microbiota in the hypertension and control groups demonstrated the dysbiosis of gut microbial in hypertension.”. See Lines 184-186.
Comment 14. Line 182: Fivefold cross-validation was repeated for five times
The reviewer is not used to seeing AUC presented as % why not AUC (95% CI) of 0.831 (0.742 to 0.919)
Response: Following the reviewer's suggestion, we have revised the AUC presentation.
See Lines 208-214.
Comment 15. The reviewer does not see the use of applying a ROC analysis to distinguish hypertension patients and hyperlipidemia from control by the abundance of bacteria genera as a simple clinical/laboratory (blood pressure, cholesterol) test can provide the classification.
Response: Thank you very much for pointing out this. This has attracted our attention. At this moment, we established a random forest model to distinguish the patients from control relying on the abundance of bacteria genera. Compared to some classic clinical/laboratory approaches, it processes promising and special advantages as a novel auxiliary measurement for diagnosis in clinic practice. The interpretation is listed below. See Lines 388-418.
For most patients with hypertension, measuring blood pressure is a common usage for hypertension diagnosis. According to the Guideline for the prevention, detection, evaluation and management of high blood pressure in adults, patients with blood pressure ≥130 mmHg and/or diastolic pressure ≥80 mmHg are diagnosed with hypertension, and recommended to receive treatment[1]. However, blood pressure is not a stable but dynamic parameter from minute to minute. Previous study reported blood pressure presented a circadian rhythm during 24-hour time with highest readings in morning and evenings and lowest readings at night[2]. Moreover, it also fluctuates in response to temperature, noise, emotional stress, consumption of food or liquid, dietary factors, physical activity, even in changes in posture, such as standing-up. For example, White-coat hypertension (WCH) is a familiar phenomenon and also an urgent tough problem to be solved for doctors. Due to the fear that blood pressure measurement may indicate future illness, subjects shows hypertension during the clinic visits resulting in isolated clinic hypertension[3,4]. All above mentioned strongly suggested measuring blood pressure is not an accurate approach for hypertension diagnosis.
On the other hand, as high-risk factor and common comorbidity, hyperlipidemia diagnosis is highly depending on blood sample[5]. Although modern technology reduces the probability of infection and transfusion transmitted diseases, invasive examination remains a challenge to medical condition in most rural area. Especially, because of concerns over pain or health care, patients are reluctant to take regular blood tests. Therefore, convenient and safe method for hyperlipidemia diagnosis is in need for people in less-developed regions.
As numerous studies proved, gut microbiota may exert a great influence on the functionality and pathophysiology of human diseases, including obesity, type II diabetes, cardiovascular diseases and some psychiatric disorders, while a deluge of research utilized it to identity patients[6]. It was reported that gut microbes could be utilized as non-invasive biomarkers for early hepatocellular carcinoma using random forest classifier[7]. Another research also demonstrated Artificial Neural Network training by relative abundance of microbiota could identify insomnia patients without subjective bias in clinic practice[8]. Thus, we incorporated machine technology to identify discriminant genera and established a prediction model for hypertension and hyperlipidemia diagnosis without disadvantages other than previous classic method, which may provide a new perspective for the interpretation of diseases and additional auxiliary diagnosis especially for rural area in the future.
Reference
[1] McManus RJ, Mant J. Hypertension: New US blood-pressure guidelines-who asked the patients? Nat Rev Cardiol 2018;15:137–8. doi:10.1016/j.jacc.2017.11.006
[2] Van Berge-Landry HM, Bovbjerg DH, James GD. Relationship between waking-sleep blood pressure and catecholamine changes in African-American and European-American women. Blood Press Monit 2008;13:257–62. doi:10.1097/MBP.0b013e3283078f45
[3] Bloomfield DA, Park A. Decoding white coat hypertension. World J Clin Cases 2017;5:82. doi:10.12998/wjcc.v5.i3.82
[4] Cai P, Peng Y, Wang Y, et al. Effect of white-coat hypertension on arterial stiffness A meta-analysis. Med (United States) 2018;97. doi:10.1097/MD.0000000000012888
[5] Ames RP. Hyperlipidemia in hypertension: causes and prevention. Am Heart J 1991;122:1219–24. doi:10.1016/0002-8703(91)90943-C
[6] Cho I, Blaser MJ. The human microbiome: At the interface of health and disease. Nat. Rev. Genet. 2012. doi:10.1038/nrg3182
[7] Ren Z, Li A, Jiang J, et al. Gut microbiome analysis as a tool towards targeted non-invasive biomarkers for early hepatocellular carcinoma. Gut 2018;1–10. doi:10.1136/gutjnl-2017-315084
[8] Xie L, Liu B, lin W, et al. Gut Microbiota as a subjective measurement for auxiliary diagnosis of insomnia disorder. Front Microbiol Published Online First: 2019. doi:10.3389/FMICB.2019.01770
Comment 16. Paragraph 3.4 is not readable, it contains too much information, already provided in the figure 4, and the main message is lost.
Response: Thank you very much for the reviewer's suggestion. We have modified and supplemented relevant contents. See Lines 244-269.
Comment 17. Discussion
The effect of medication intake should be better discussed
Response: Thank you very much for the reviewer's suggestion. We have supplemented relevant content. See Lines 85-87 and 381-387.
Comment 18. P8 line 302: what is the relevance of Prevotella, originating from mouth and vagina,…?
Response: Thank you very much for the reviewer's comments. We have revised the sentence as “Prevotella was found to be more abundant in the AH and HLD groups than in the NH and control groups in this study” See Line 315-316.
For English language
Response: This paper was edited by Elsevier Language Editing Services.
Reviewer 2 Report
The authors present the main outcomes of the main genera populating the gut of hypertensive patients and hyperlipidemia individuals.
The paper is well written and clear. The study has some limitations which are however acknowledged in the paper.
Just a few points to address:
Il. 139--141 please, delete.
Figure 1, 2 and 4. More visible
L.217 replace "et al."
Capital letter after full stop in lines273 and 350
Line 300--350 please divide into paragraphs
Lines 391--393 delete, please.
Author Response
Comment 1. Il. 139--141 please, delete.
Response: Thank you very much for the reviewer's suggestion. We have deleted the sentence of “, and subsequently visualized with the R ggplot2 package”. See Line 143-145.
Comment 2. Figure 1, 2 and 4. More visible
Response: Thank you very much for the reviewer's suggestion. We have increased the pixel size of all the figures.
Comment 3. L.217 replace "et al."
Response: Thank you very much for the reviewer's suggestion. We have replaced “et al.”
Comment 4. Capital letter after full stop in lines273 and 350
Response: Thank you very much for the reviewer's suggestion. We have checked all the sentences in lines 244 and 351.
Comment 5. Line 300--350 please divide into paragraphs
Response: Thank you very much for the reviewer's suggestion. We have divided this part in three paragraphs. See Lines 309-351.
Comment 6. Lines 391--393 delete, please.
Response: Thank you very much for the reviewer's suggestion. We have deleted this part. See Line 429.
Round 2
Reviewer 1 Report
1) please add reference (Liu X et al.) for the use of the questionaire to the paper.
2) please still adapt table 1 omitting the columns with the P-values and only presenting the statisitcs as symbols next to the respective charactersitics
NH AH HLD control
e.g. BMI 27.0±3.6* 26.6±3.1* 26.5±3.0 25.3±2.9
*P<0.05 vs control
Author Response
Comment 1. please add reference (Liu X et al.) for the use of the questionaire to the paper.
Response: Thank you very much for your comments and suggestion. The reference (Liu X et al.) has been added. See Lines 114-115 and Lines 484-486.
Comment 2. please still adapt table 1 omitting the columns with the P-values and only presenting the statisitcs as symbols next to the respective characteristics.
Response: Thank you very much for the reviewer's suggestion. We have revised the contents. See Lines 165-166, Table 1.